# Standardized *Aronia melanocarpa* Extract as Novel Supplement against Metabolic Syndrome: A Rat Model

**DOI:** 10.3390/ijms20010006

**Published:** 2018-12-20

**Authors:** Vladimir JAKOVLJEVIC, Petar Milic, Jovana Bradic, Jovana Jeremic, Vladimir Zivkovic, Ivan Srejovic, Tamara Nikolic Turnic, Isidora Milosavljevic, Nevena Jeremic, Sergey Bolevich, Milica Labudovic Borovic, Miroslav Mitrovic, Vesna Vucic

**Affiliations:** 1Department of Physiology, Faculty of Medical Sciences, University of Kragujevac, Svetozara Markovica 69, 34000 Kragujevac, Serbia; vladimirziv@gmail.com (V.Z.); ivan_srejovic@hotmail.com (I.S.); 2Department of Human Pathology, 1st Moscow State Medical, University IM Sechenov, Trubetskaya street 8, Moscow 119991, Russia; bolevich2011@yandex.ru; 3Department of Pharmacy, High Medical School of Professional Studies in Cuprija, Lole Ribara 1/2, 35000 Cuprija, Serbia; milic.milic@gmail.com; 4Department of Pharmacy, Faculty of Medical Sciences, University of Kragujevac, Svetozara Markovica 69, 34000 Kragujevac, Serbia; jovanabradickg@gmail.com (J.B.); jovana.jeremic@medf.kg.ac.rs (J.J.); tamara.nikolic@medf.kg.ac.rs (T.N.T.); isidora.stojic@medf.kg.ac.rs (I.M.); nbarudzic@hotmail.com (N.J.); 5Institute of Histology and Embryology “Aleksandar Dj. Kostic”, Faculty of Medicine, University of Belgrade, Dr Subotic 8, 11000 Belgrad, Serbia; milica.borovic@supremalab.rs; 6Pharmanova, Generala Anrija 6, 11010 Belgrade, Serbia; miroslav.mitrovic@pharmanova.com; 7Institute for Medical Research, Centre of Research Excellence in Nutrition and Metabolism, University of Belgrade, Tadeusa Koscuska 1, 11129 Belgrade, Serbia; vesna.vucic.imr@gmail.com

**Keywords:** metabolic syndrome, *Aronia melanocarpa*, standardized extract, dietary strategies, supplementation

## Abstract

The aim of our study was to examine the effects of different dietary strategies, high-fat (HFd) or standard diet (Sd) alone or in combination with standardized oral supplementation (0.45 mL/kg/day) of Aronia melanocarpa extract (SAE) in rats with metabolic syndrome (MetS). SAE is an official product of pharmaceutical company Pharmanova (Belgrade, Serbia); however, the procedure for extraction was done by EU-Chem company (Belgrade, Serbia). Rats were divided randomly into six groups: control with Sd, control with Sd and SAE, MetS with HFd, MetS with HFd and SAE, MetS with Sd and MetS with Sd and SAE during 4 weeks. At the end of the 4-week protocol, cardiac function and liver morphology were assessed, while in the blood samples glucose, insulin, iron levels and systemic redox state were determined. Our results demonstrated that SAE had the ability to lower blood pressure and exert benefits on in vivo and ex vivo heart function. Moreover, SAE improved glucose tolerance, attenuated pathological liver alterations and oxidative stress present in MetS. Obtained beneficial effects of SAE were more prominent in combination with changing dietary habits. Promising potential of SAE supplementation alone or in combination with different dietary protocols in triggering cardioprotection should be further examined in future.

## 1. Introduction

Metabolic syndrome (MetS) represents one of the metabolic disorders characterized by abdominal obesity, dyslipidemia, hypertension, insulin resistance and diabetes mellitus (DM) type 2 [1,2]. There is a great concern since MetS directly promotes the development of cardiovascular disorders, possibly because it results in increased oxidative stress and low-grade inflammation [3]. Therapeutic approaches involve dietary restriction or a combination of synthetic antidiabetic and hypolipidemic drugs [4]. However, increasing incidence of MetS associated with the undesirable side—effects and high cost of available drugs indicates the need to discover new, less harmful herbal medicines efficient in controlling both blood glucose and lipids [4,5]. Therefore, a lot of plant extracts as well as plant-derived biomoleculs such as polyphenols, have been under research for the prevention and therapy of MetS [1]. It has been well documented that polyphenols, especially anthocyanins and quercetin, exert the potential to enhance the glucose uptake by muscle and adipocyte cells, thus exerting antidiabetic effect [6].

*Aronia melanocarpa* (*A. melanocarpa*) or black chokeberry is a fruit/plant which belongs to the *Rosaceae* family and is native to North America [7]. However, it has been commonly used in Europe as ingredient for juices, wine, jams, teas and cordial liqueurs [8,9]. *A. melanocarpa* represents one of the richest sources of polyphenols among fruits, with anthocyanins and flavonoids identified as major components responsible for its therapeutic potential [10,11]. Recent researches have focused attention on *A. melanocarpa* due to its numerous health benefits in a broad range of pathological conditions [12]. It has been reported that fruit and extracts of *A. melanocarpa* exert gastroprotective, hepatoprotective, antiinflammatory and antiproliferative activity [12,13]. Furthermore, the health-promoting effects of extracts of this plant involve antiatherosclerotic, antiplatelet and hypoglycemic properties [7,14]. Moreover, it was previously confirmed that *A. melanocarpa* extract may reduce systolic and diastolic pressure and be useful in the management of DM [5,13]. However, to our best knowledge, the effectiveness of *A. melanocarpa* extract in combination with different diet regimens in the treatment of MetS has been not investigated so far.

Therefore, the aim of our study was to examine different dietary strategies, high-fat (HFd) or standard diet (Sd) alone or in combination with standardized *A. melanocarpa* extract (SAE) supplementation, and their potential benefits in the prevention and treatment of various complications in rats with MetS.

## 2. Results

### 2.1. Body Mass Index (BMI) of Healthy and Rats with MetS after 4-Weeks of Dietary Changes

BMI was statistically higher in groups with MetS (МetS + HFd, МetS + HFd + SAE, МetS + Sd, МetS + Sd + SAE) than in healthy rats (CTRL, SAE). However, 4 weeks after dietary regime, МetS + Sd and МetS + Sd + SAE groups had significantly lower BMI levels than МetS + HFd and МetS + HFd + SAE, while МetS + Sd + SAE had significantly lower BMI level than МetS + Sd (Figure 1).

### 2.2. Changes in Blood Pressure and Heart Rate in Healthy and Rats with MetS on the Different Diet Regime

Systolic and diastolic blood pressure (SBP and DBP), as well as heart rate (HR) were increased in rats with MetS compared to healthy rats, as excepted (Figure 2A–C). More interesting was that with the addition of SAE and Sd in rats with MetS, SBP was statistically decreased compared to MetS + Sd and MetS + HFd + SAE (Figure 2A). DBP was significantly increased in MetS + HFd in comparison with MetS + HFd + SAE, MetS + Sd and MetS + Sd + SAE, while in MetS + Sd + SAE group DBP was significantly lower than in other MetS groups (Figure 2B). SAE treatment in MetS + HFd + SAE group induced significant increase of HR compared to MetS + HFd and MetS + Sd groups (Figure 2C).

### 2.3. Effect of Dietary Changes in Healthy and Rats with MetS in In Vivo Cardiac Function

SAE supplementation in healthy rats significantly increased interventricular septal wall thickness at end diastole (IVSd), left ventricle posterior wall thickness at end diastole (LVPWd), interventricular septal wall thickness at end systole (IVSs), left ventricle posterior wall thickness at end systole (LVPWs), fractional shortening (FS) and reduced left ventricle internal dimension at end systole (LVIDs) and left ventricle internal dimension at end diastole (LVIDd) compared to control. On the other hand, IVSd and IVSs were significantly decreased while LVIDd was significantly increased in MetS + HFd compared to SAE group. More importantly, FS was statistically decreased in MetS + HFd compared to CTRL and SAE, as well as in MetS + HFd + SAE and MetS + Sd compared to SAE. SAE supplementation in rats with MetS fed with Sd significantly increased LVIDd, IVSs and LVIDs compared to MetS + Sd group (Table 1).

### 2.4. Effect of Dietary Changes in Healthy and Rats with MetS on Ex Vivo Cardiac Function

Figure 3 shows the values of ex vivo measured cardiac function parameters and coronary flow, during pressure changing protocols (PCPs) on the Langendorff apparatus. To examine the potential difference due to various dietary habits, we compared the percentage of decrease (−) or increase (+) between PCP 1 and PCP 2 in the group (Table 2).

Major changes in maximum rate of pressure development in the left ventricle (dp/dt max) in PCP 1 versus PCP 2 were observed at coronary perfusion pressure (CPP) = 60 cm H_2_O and 80 cm H_2_O in MetS + HFd (−24.77; −35.3) and MetS + Sd (−24.23; −28.11) groups. In MetS groups that were fed with combination of mentioned diets and SAE, this parameter was not changed during pressure changing protocols. On the other hand, the most significant differences in minimum rate of pressure development in the left ventricle (dp/dt min) were observed in CTRL group at CPP = 60 cm H_2_O, 80 cm H_2_O and 100 cm H_2_O (−21.25; −22.03; −29.7), while during the PCPs there were no significant changes in systolic left ventricular pressure (SLVP), HR, and coronary flow (CF) in any of the examined groups (Figure 3, Table 2).

### 2.5. Effect of Dietary Changes in Healthy and Rats with MetS on Glucose and Insulin Levels during Oral Glucose Tolerance Test (OGTT)

#### 2.5.1. Effects on Glucose Levels during OGTT

The average blood glucose values during an OGTT were present in Figure 4. Fasting blood glucose concentrations were significantly increased in all MetS groups compared to healthy groups, except in МetS + HFd + SAE where glucose level was the lowest among the MetS groups. In addition, glucose level was lower in МetS + Sd + SAE than in МetS + Sd. The similar trend was maintained during 30, 60 and 120 min, while in 180′ the highest level was in МetS + HFd.

#### 2.5.2. Effects on Insulin Levels during OGTT

Table 3 shows the insulin concentration measured during the OGTT. The SAE group had the lowest, while МetS + Sd group had the highest insulin concentration measured fasting (0′), as well as 3 h after glucose administration (180′). In MetS groups fasting insulin concentration was significantly higher than in CTRL group. Moreover, insulin concentration was significantly lower in МetS + Sd + SAE than in МetS + Sd group, in both measured moments of interest.

### 2.6. Effect of Dietary Changes in Healthy and Rats with MetS on Serum Iron Levels

SAE supplementation significantly increased iron levels in serum of healthy and rats with MetS fed with HFd or Sd than in non-treated groups. CTRL group had the lowest values of iron in relation to all other examined groups (Figure 5).

### 2.7. Evaluation of Systemic Redox State

Level of nitrites (NO_2_^−^) was significantly decreased in SAE group compared to CTRL group, and significantly increased in MetS groups compare to CTRL and SAE groups. The highest values of NO_2_^−^ were observed in MetS + HFd group, while with the addition of SAE in HFd or Sd these values drop dramatically. Interestingly, MetS + Sd had significantly higher NO_2_^−^ levels compare to MetS + HFd + SAE group (Figure 6A).

The highest level of superoxide anion radical (O_2_^−^) was noticed in MetS groups untreated with SAE extract. Moreover, SAE supplementation in healthy and rats with MetS significantly reduced O_2_^−^ levels (Figure 6B).

Hydrogen peroxide (H_2_O_2_) levels were significantly increased in all examined group (except in MetS + HFd + SAE) compared to CTRL (Figure 6C).

Superoxide dismutase (SOD) activity was significantly reduced in rats with MetS, compare to healthy rats. The transition to a standard food with or without SAE supplementation led to a significant increase of SOD activity (Figure 6D).

Catalase (CAT) activity was significantly higher in SAE group compared to other examined groups, except MetS + Sd + SAE. On the other hand, in CTRL group this parameter was significantly increased compared to MetS + HFd and MetS + HFd + SAE and significantly decreased compared to MetS + Sd + SAE. With the addition of SAE in diet of rats with MetS, we observed significant increasement of CAT activity in comparison to MetS rats untreated with SAE (Figure 6E).

Reduced glutathione (GSH) levels were significantly increased in SAE and MetS + HFd + SAE compared to other observed groups (Figure 6F).

### 2.8. Histological Analysis of Liver Tissue

As shown in Figure 7, in CTRL, SAE, МetS + HFd + SAE, МetS + Sd and МetS + Sd + SAE liver tissue is a common feature. Liver lobulus is fully preserved. Hepatocytes are correctly arranged in the liver plates, without change. Sinusoidal capillaries are common characteristics, also without change. Fibrosis as well as inflammation have not been detected. On the other hand, in МetS + HFd there was a microvesicular steatosis (fatty change).

## 3. Discussion

Several epidemiologic studies have implicated visceral fat as a major risk factor for insulin resistance, type 2 diabetes mellitus, cardiovascular disease, stroke, metabolic syndrome and death [15]. Taking into consideration increasing incidence of MetS and its related complications we wanted to estimate the effectiveness of HFd cessation and introduction of polyphenol-rich extract (SAE) on weight gain, cardiac function, glucose tolerance, serum insulin and iron levels, as well as systemic redox state and morphological characteristic of the liver.

Our results clearly show that the highest increase in BMI was observed in MetS + HFd, which was expected as many data suggest weight gain in rats exposed to HFd for different periods [16,17,18]. Introduction of Aronia extract in MetS + HFd group suppressed the body weight gain and decreased BMI; however changing dietary habits from high-fat to standard food had better anti-obesity effect when compared to MetS + HFd group. Moreover, the most prominent reduction in body weight and BMI was achieved by standard diet associated with consumption of SAE extract.

Taking into account, that cardiovascular complications in MetS are very common, we wanted to examine cardiovascular effects such as their ability to affect blood pressure as well as in vivo and ex vivo cardiac function, after dietary changes. Transition from a high caloric to normal fat diet-induced a decline in diastolic blood pressure. Nevertheless, the highest hypotensive effect in rats with MetS, evidenced with a drop in both systolic and diastolic pressure was reached when this regimen was combined with Aronia extract. Extract of Aronia was able to induce a drop in diastolic pressure even in rats who were fed with high-fat food continuously. On the contrary, in healthy rats SAE did not affect blood pressure. This is in line with the data that both Aronia berries and Aronia polyphenol extracts reduce quite effectively both SBP and DBP in spontaneously hypertensive rats [19]. The proposed mechanism might be through inhibition of the kidney renin-angiotensin system [20]. Others also found that *A. melanocarpa* extract decreases blood pressure in experimental model of hypertension [21,22]. Growing evidence suggests that the flavonoid-rich foods intake is related with decline in SBP and DBP, so we assume that the blood pressure-lowering effect of SAE is attributed to polyphenols [22,23]. Hypotensive effect was confirmed in patients with metabolic syndrome as well [24].

During blood pressure measuring, HR was also registered. Increase in HR in groups with MetS compared to healthy rats may be explained by an increase in sympathetic nervous system activity induced by HFd [25]. In that sense obtained decrease in HR after the transition to standard diet in MetS + Sd group appears to be logical. However, there was no change in HR when SAE was added to a dietary regimen in rats with MetS who were on a Sd, while we noticed an increase in rats on HFd. Moreover, Aronia extract did not alter heart rate in healthy rats, reflecting preserved myocardial function and contractility. Similar results were found in previous investigations regarding the effects of polyphenol-containing extracts on HR [26].

An echocardiographic examination illustrated that the highest impact of Aronia extract on in vivo myocardial functions was found in healthy rats, where we observed significant increase in IVSd, LVPWd, IVSs, LVPWs, FS and decrease in LVIDs compared to CTRL. On the other hand, addition of Aronia extract in rats with MetS during both HFd and Sd did not significantly affect cardiac function compared to MetS + HFd group. The greatest benefit of SAE involves improvement in systolic function, manifested as a significant increase in fractional shortening (FS) in healthy rats relative to almost all other groups. Similar values in healthy and MetS + Sd + SAE and MetS + HFd + SAE group suggest that Aronia extract was capable of improving fractional shortening during both high-calorie and standard dietary conditions in the presence of metabolic syndrome. In accordance with our findings, it has been previously reported that polyphenolics and plants rich in polyphenolics had effect in lessening the pathological alterations in FS promoted by MetS [27,28,29]. A decrease in systemic blood pressure after Aronia extract treatment may increase fractional shortenings, resulting in increased myocardial contractility [30].

Similar results were obtained during ex vivo, retrograde perfusion on Langendorff. Cardiac contractility, estimated by maximum and minimum rate of left ventricle pressure development, (dp/dt mSAE and dp/dt min), was preserved in MetS groups treated with SAE in combination with HFd or Sd, especially in normoxic conditions (CPP = 60 and 80 cm H_2_O). Furthermore, addition of Aronia extract in healthy rats significantly improved heart relaxation (for the CPP values from 60–100 cm H_2_O) compared to CTRL group. These results confirm the assumption that Aronia extract triggers cardioprotection, most probably because of its antioxidant, antiinflammatory, vasorelaxant and antithrombotic effects [31].

To estimate if SAE extract might improve glucose tolerance, which is strongly related to insulin resistance and insulin secretion, we performed oral glucose tolerance (OGT) test. Our results highlighted that SAE extract did not affect fasting glucose level in healthy rats, while it exerted hypoglycemic effect in animals with MetS on both HFd and Sd. The similar trend was noticed during 30, 60, 120 and 180 min, except the fact that in 120 and 180 min there was no difference in glucose level between MetS + Sd and MetS + Sd + SAE. Regarding the concentration of insulin, it was the lower in group of healthy rats receiving SAE group compared to healthy untreated group in 0′. Moreover, higher insulin concentration in MetS + HFd group wasn’t diminished after adding SAE extract. On the other hand, this extract potentiates effect of standard food on lowering insulin, as evidenced by a decrease in insulin concentration in MetS + Sd + SAE compared to MetS + Sd group in 0′ and 180′. In line with our observation, better glucose tolerance achieved by treatment with SAE extract was confirmed by several papers. Other authors showed the beneficial effects of *A. melanocarpa* extract on attenuating insulin resistance and improving insulin sensitivity in HFd-induced obese mice [32]. Furthermore, glucose lowering potential was confirmed in patients with DM as well [33]. Proposed mechanism though which Aronia exerts hypoglicemic effect involve inhibition of dipeptidyl peptidase IV and α-glucosidase activities [34].

Since iron is an essential trace element that has been involved in maintenance of regular homeostasis, understanding the influence of SAE alone or in combination with different dietary regiments its serum levels would be of a great importance [35]. We found the highest level of iron in group of healthy animals receiving SAE, while increased level of iron was found in all groups with MetS compared to healthy rats. Previous study indicates that increased iron stores have been associated with MetS [36]. Changing dietary habits from high-fat to standard food did not result in change of iron, while addition of SAE in this group induced significant rise when compared to MetS group. This is in agreement with other research which showed that supplementation with Aronia juice increased serum level of iron [34]. In fact, certain flavonoids have potential to chelate iron and decrease iron absorption through mechanism independent of the hepcidin, a hormone included in iron homeostasis [35,37]. On the other hand, other flavonoids may decrease the activity of hepcidin resulting in increased iron uptake and serum iron levels, which may be an explanation for the SAE extract-induced increase in our research [35,38].

Increased oxidative stress has been linked with pathogenesis of MetS, thus indicating the need for consuming natural antioxidants from food sources in treatment of MetS-related diseases [39]. In that sense, in order to test if 4-week SAE supplementation alters systemic redox homeostasis we determined levels of pro-oxidants, as well as capacity of antioxidant defense system in blood samples. Our results demonstrated that SAE consumption led to a drop in NO_2_^−^ and O_2_^−^ and rise in CAT and GSH in healthy rats. Increase in CAT activity which catalyzes the decomposition of hydrogen peroxide to water and oxygen support unchanged values of H_2_O_2_. As it was expected increased generation of pro-oxidants and decreased activity of antioxidant enzymes SOD and CAT were noticed in MetS + HFd group compared to control. Introduction of three different dietary strategies such as consumption of SAE or transition to standard diet or its combination induced decline only in NO_2_^−^ compared to MetS + HFd. Regarding the antioxidant status, the highest impact of increase in activities of antioxidant enzymes was noticed when standard diet was combined with SAE treatment. Striking evidence indicate that polyphenols might increase antioxidant capacity via rise in activities of SOD, CAT and GSH-peroxidase and act as direct free radical scavengers as well [40,41]. Chelation of iron ions which catalyze several free radical-generating reactions is one of the mechanisms underlying antioxidant effects of polyphenols. Nevertheless, rise in iron level induced by SAE in our research lead us to a hypothesis that enhanced activity of antioxidant enzymes and direct scavenging rather than iron chelation were responsible for antioxidant potential of applied extract. However, poor bioavailability of polyphenols through food intake suggests necessity for polyphenol-enriched foods or supplements treatment such as our extract [42].

MetS in combination with high-fat altered structure of liver tissue manifested as microvesicular hepatic steatosis. Nevertheless, transition from high-fat to standard food and combined approach which involved SAE extract consumption associated with both dietary protocols significantly normalized liver changes in MetS groups. Obtained positive effects in those groups are evidenced by the absence of fibrosis and inflammation. Ability of anthocyanins in Aronia to diminish liver steatosis induced by MetS was documented before, so we may hypothesize that these bioactive compounds are responsible for the beneficial effects in our study [43,44]. Some results show beneficial effects of *A. melanocarpa* against hepatic lipid accumulation through the inhibition of peroxisome proliferator-activated receptor γ2 (PPARγ2) expression along with improvements in body weight, liver functions, lipid profiles and antioxidant capacity suggesting the potential therapeutic efficacy of *A. melanocarpa* on nonalcoholic fatty liver disease [45]. Recently, it was showed clearly an increase in acetylcholinesterase (AChE) and butyryl cholinesterase activity and disruption of lipid metabolism in patients with MetS. After supplementation of MetS patients with *A. melanocarpa* extract, a decrease in AChE activity and oxidative stress was noted [46].

## 4. Material and Methods

### 4.1. Ethical Approval

This research was carried out in the laboratory for cardiovascular physiology of the Faculty of Medical Sciences, University of Kragujevac, Serbia. The study protocol was approved (number: 119-01-5/14/2017-09, date: 30 June 2017) by the Ethical Committee for the welfare of experimental animals of the Faculty of Medical Sciences, University of Kragujevac, Serbia. All experiments were performed according to EU Directive for welfare of laboratory animals (86/609/EEC) and principles of Good Laboratory Practice.

### 4.2. Animals

Sixty *Wistar albino* rats (males, six weeks old, body-weight 200 ± 30 g, on beginning of experiments) were included in the study. They were housed at temperature of 22 ± 2 °C, with 12 h of automatic illumination daily. The rats were randomly divided into two groups: healthy animals (*n* = 20), fed with Sd which contains 9% fat, 20% protein, 53% starch, 5% fiber and animals with MetS (*n* = 40), fed with HFd which contains 25% fat, 15% protein, 51% starch and 5% fiber during 4 weeks. After one month on their respective diets, rats from MetS group after 6–8 h of starvation received one dose of streptozotocin intraperitoneally. Streptozotocin was prepared *ex tempore* by dissolving in citrate buffer and, depending on the body weight, it was administered in a dose of 25 mg/kg body weight [47]. Three days (72 h) after streptozotocin treatment e and 12 h after starvation fasting glucose and insulin level as well as blood pressure, were measured. Animals with systolic blood pressure greater than 140 mmHg, diastolic blood pressure more than 85 mmHg, fasting glucose level above 7.0 mmol/L and fasting insulin level over 150 µLU/mL were included in the study and were used in the study as rats with MetS.

Healthy and rats with MetS were divided into 6 groups as follows: CTRL—healthy rats, fed with a Sd for 4 weeks; SAE—healthy rats, fed with a Sd and treated with highly concentrated Aronia extract standardized with polyphenol content—SAE in the dose 0.45 mL/kg/day per os for 4 weeks; MetS + HFd—rats with MetS, fed with HFd for 4 weeks; MetS + HFd + SAE—rats with MetS, fed with HFd and treated with SAE (0.45 mL/kg/day, per os) for 4 weeks; MetS + Sd—rats with MetS, fed with a Sd for 4 weeks; MetS + Sd + SAE—rats with MetS, fed with a Sd for 4 weeks and treated with SAE (0.45 mL/kg/day, per os) for 4 weeks.

Standardized Aronia extract (SAE) is official product of pharmaceutical company Pharmanova (Belgrade, Serbia); however procedure of extraction was done by EU-Chem company (Belgrade, Serbia).

### 4.3. Measurement of BMI

At the end of the study protocol, body weight and body length were measured. Body length represents nose-anus length. Those parameters were used to calculate the BMI of the rats as follows: 
Body mass index (BMI) = body weight (g)/length^2^ (cm^2^)
(1)

### 4.4. Evaluation of Blood Pressure and Heart Rate

A day before sacrificing animals, the blood pressure and heart rate were measured by a tail-cuff noninvasive method BP system (Rat Tail Cuff Method Blood Pressure Systems (MRBP-R), IITC Life Science Inc., Los Angeles, CA, USA) [48].

### 4.5. Evaluation of in vivo Cardiac Function

After accomplishing 4-week treatment, transthoracic echocardiograms were performed. Mixture of ketamine—50 mg/kg and xylazine—10 mg/kg intraperitoneally was used as anesthesia. Echocardiograms were performed using a Hewlett-Packard Sonos 5500 (Andover, MA, USA) sector scanner equipped with a 15.0-MHz phased-array transducer as previously described [49]. From the parasternal long-axis view in 2-dimensional mode, and M-mode cursor was positioned perpendicularly to the interventricular septum and posterior wall of the left ventricle (LV) at the level of the papillary muscles and M-mode images were obtained. Interventricular septal wall thickness at end diastole (IVSd), LV internal dimension at end diastole (LVIDd), LV posterior wall thickness at end diastole (LVPWd), interventricular septal wall thickness at end systole (IVSs), LV internal diameter at end systole (LVIDs) and LV posterior wall thickness at end systole (LVPWs) were recorded with M-mode. Fractional shortening percentage (FS%) was calculated from the M-mode LV diameters using the equation: [(LVEDd−LVESd)/LVEDd] × 100%. Where LVEDd is left ventricular end diastolic diameter and LVESd is left ventricular end systolic diameter.

### 4.6. Evaluation of ex vivo Cardiac Function

Following 4-week protocol, after short-term narcosis induced by intraperitoneal application of ketamine (10 mg/kg) and xylazine (5 mg/kg) and premedication with heparin as an anticoagulant, animals were sacrificed by decapitation. Then the chest was opened via midline thoracotomy, hearts were immediately removed and immersed in cold saline and aortas were cannulated and retrogradely perfused according to *Langendorff* technique, under gradually increasing coronary perfusion pressure (CPP) from 40 to 120 cm H_2_O [50]. The composition of Krebs-Henseleit buffer used for retrograde perfusion was as follows (mmol/L): NaCl 118 mmol/L, KCl 4.7 mmol/L, MgSO_4_ × 7H_2_O 1.7 mmol/L, NaHCO_3_ 25 mmol/L, KH_2_PO_4_ 1.2 mmol/L, CaCl_2_ × 2H_2_O 2.5 mmol/L, glucose 11 mmol/L, pyruvate 2 mmol/L, equilibrated with 95% O_2_ plus 5% CO_2_ and warmed to 37 °C (pH 7.4).

After placing the sensor (transducer BS473-0184, Experimetria Ltd., Budapest, Hungary) in the left ventricle, the following parameters of myocardial function have been measured: maximum rate of pressure development in the left ventricle (dp/dt max), minimum rate of pressure development in the left ventricle (dp/dt min), systolic left ventricular pressure (SLVP), diastolic left ventricular pressure (DLVP), heart rate (HR). Coronary flow (CF) was measured flowmetrically. Following the establishment of heart perfusion, the hearts were stabilized within 30 min with a basal coronary perfusion pressure of 70 cm H_2_O. To examine the heart function, after stabilization period, the perfusion pressure was gradually decreased to 60, and then increased to 80, 100 and 120 cm H_2_O and reduced to 40 cm H_2_O (pressure changing protocol 1, PCP 1) and again gradually increased from 40 to 120 cm H_2_O (pressure changing protocol 1, PCP 1).

### 4.7. Oral Glucose Tolerance Test

Oral glucose tolerance test (OGTT) was performed at the end of 4-week protocol and a day before sacrificing animals. After overnight (12–14h) fasted animals, the blood sample was taken by tail bleeding to determine the fasting blood glucose and insulin level (0 min) and then glucose was administered orally in a dose of 2 g/kg body weight and blood samples were taken at 30, 60, 120 and 180 min after glucose loading. Glucose levels were determined in 0, 30, 60, 120 and 180 min, using glucometer (Accu-Chek, Roche Diagnostics, Indianapolis, IN, USA) with its corresponding strips. At 0 and 180 min, insulin levels were assessed in plasma samples by the enzyme-linked immunosorbent assay (ELISA) method as previously described [51].

### 4.8. Evaluation of Serum Iron Levels and Systemic Redox State

In the moment of sacrificing animals blood samples were collected from jugular vein in order to estimate serum iron levels and systemic oxidative stress response. The levels of serum iron (SI) was determined on a biochemical analyzer (Dimension, Dade Behring, Milton Keynes, UK, USA) and the results were expressed in μg/L.

In plasma the following pro-oxidants were determined: the levels of nitrites (NO_2_^−^), superoxide anion radical (O_2_^−^) and hydrogen peroxide (H_2_O_2_). Parameters of antioxidative defence system, such as activities of superoxide dismutase (SOD) and catalase (CAT) and level of reduced glutathione (GSH) were determined in erythrocytes samples.

NO_2_^−^ was determined as an index of NO production with Griess reagent. The method for detection nitrate in plasma is based on the Green and coworkers proposal and measured spectrophotometrically at a wavelength of 543 nm [52]. The concentration of O_2_^−^ in plasma was measured at 530 nm, after the reaction of nitro blue tetrazolium in TRIS buffer [53]. The measurement of H_2_O_2_ is based on the oxidation of phenol red by hydrogen peroxide, in a reaction catalyzed by horseradish peroxidase (HRPO) as previously described by Pick and colleagues. The level of H_2_O_2_ was measured at 610 nm [54].

For determination of antioxidant parameters, isolated erytrocytes were prepared according to McCord and Fridovich [54]. CAT activity were determined at 360 nm toward to Beutler [55]. Lysates were diluted with distilled water (1:7 *v*/*v*) and treated with chloroform-ethanol (0.6:1 *v*/*v*) to remove hemoglobin [51]. SOD activity was determined by the epinephrine method of Misra and Fridovich. Detection was performed at 470 nm [56]. Level of GSH is based on GSH oxidation via 5,5-dithiobis-6,2-nitrobenzoic acid as previously described by Beutler. Measuring was performed at 420 nm [57].

### 4.9. Histological Analysis of Liver Tissue

Liver tissue samples were fixed in 4% buffered paraformaldehyde solution and immersed in paraffin. Afterwards 4-micrometre-thick sections were stained with hematoxylin and eosin (H&E) [43].

### 4.10. Statistical Analysis

IBM SPSS Statistics 20.0 Desktop for Windows was used for statistical analysis. Distribution of data was checked by Shapiro–Wilk test. Where distribution between groups was normal, statistical comparisons were performed using the one-way analysis of variance (ANOVA) tests with a Tukey’s post hoc test for multiple comparisons. Kruskal–Wallis was used for comparison between groups where the distribution of data was different than normal. Values of *p* < 0.05 were considered to be statistically significant.

## 5. Conclusions

Our results highlighted cardioprotective potential of SAE in treatment of MetS, involving lowering blood pressure and favorable effects on heart function. Furthermore, SAE effectively suppressed the body weight gain, improved glucose tolerance and attenuated liver steatosis and oxidative stress present in MetS, thus indicating its promising role in management of MetS-related diseases. Moreover, increase in iron concentration indicates its health benefits in iron deficiency. Obtained beneficial effects would be more prominent in c combination with changing dietary habits. This research may be a starting point for further experimental and clinical investigations which would fully evaluate the effects of SAE alone or in combination with different dietary protocols in various models of chronic diseases.

## Figures and Tables

**Figure 1 ijms-20-00006-f001:**
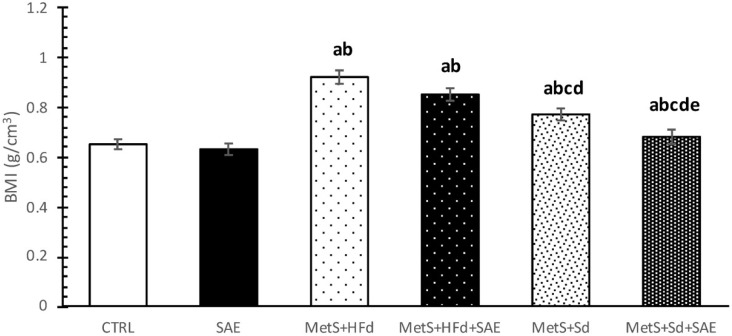
BMI in examined groups 4 weeks after dietary changes. Values are expressed as mean ± standard deviation for 10 animals, for each group. For statistical significance were considered values *p* < 0.05. ^a^ Statistical significance in relation to control (CTRL) group; ^b^ Statistical significance in relation to standardized *A. melanocarpa* extract (SAE) group; ^c^ Statistical significance in relation to МetS + HFd group; ^d^ Statistical significance in relation to МetS + HFd + SAE group; ^e^ Statistical significance in relation to МetS + Sd group.

**Figure 2 ijms-20-00006-f002:**
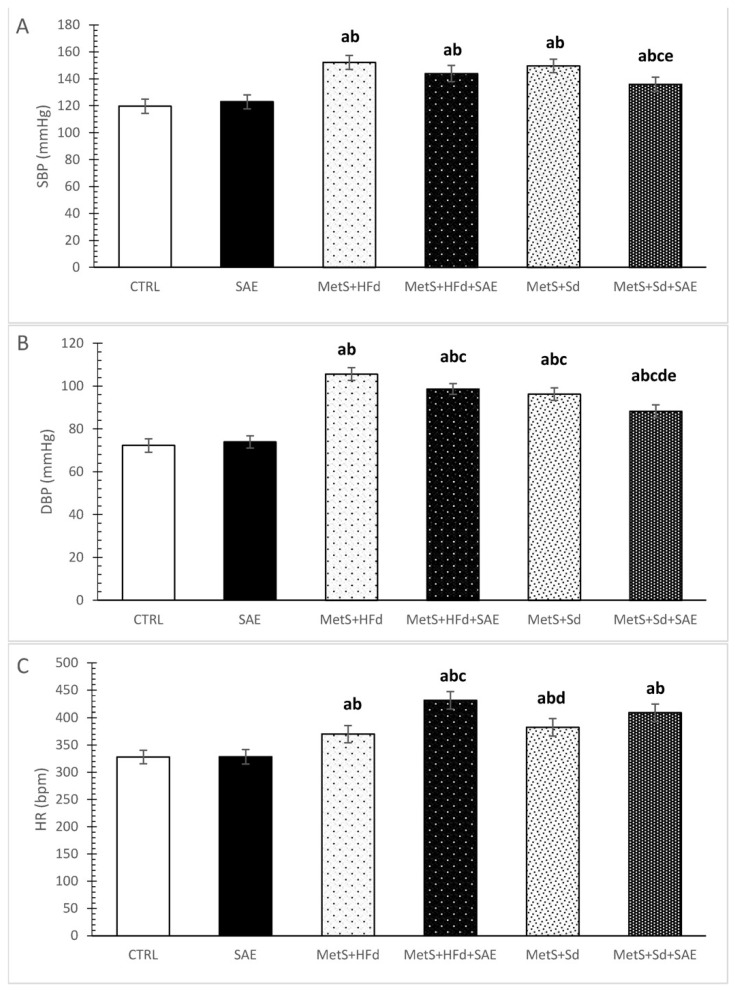
Changes in blood pressure and heart rate in healthy and rats with MetS on the different diet regime: (**A**) systolic blood pressure (SBP, mmHg); (**B**) diastolic blood pressure (DBP, mmHg); (**C**) heart rate (HR, bpm). Values are expressed as mean ± standard deviation for 10 animals, for each group. For statistical significance were considered values *p* < 0.05. ^a^ Statistical significance in relation to CTRL group; ^b^ Statistical significance in relation to SAE group; ^c^ Statistical significance in relation to МetS + HFd group; ^d^ Statistical significance in relation to МetS + HFd + SAE group; ^e^ Statistical significance in relation to МetS + Sd group.

**Figure 3 ijms-20-00006-f003:**
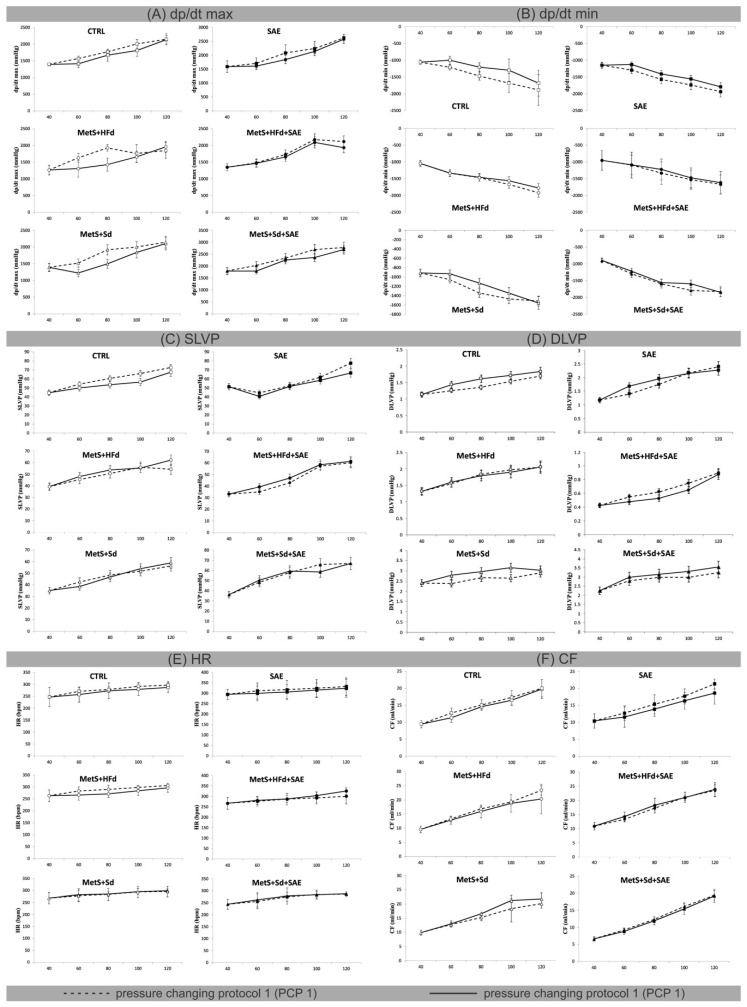
Effect of dietary changes in healthy and rats with MetS on cardiac function parameters, measured ex vivo: (**A**) values of dp/dt max within each of 6 groups during pressure changing protocol 1 (PCP 1) and pressure changing protocol 2 (PCP 2); (**B**) values of dp/dt min within each of 6 groups during PCP 1 and PCP 2; (**C**) values of SLVP within each of 6 groups during PCP 1 and PCP 2; (**D**) values of diastolic left ventricular pressure (DLVP) within each of 6 groups during PCP 1 and PCP 2; (**E**) values of HR within each of 6 groups during PCP 1 and PCP 2; (**F**) values of CF within each of 6 groups during PCP 1 and PCP 2. All values are expressed as mean ± standard deviation for each group.

**Figure 4 ijms-20-00006-f004:**
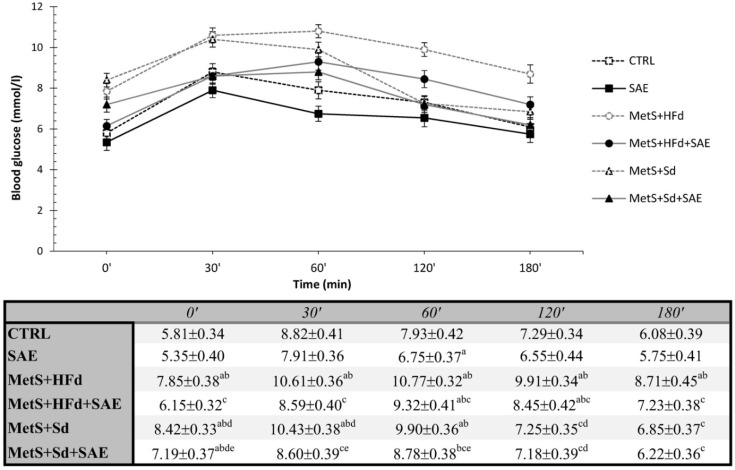
Effect of dietary changes in healthy and rats with MetS on glucose levels during OGTT. Values are expressed as mean ± standard deviation for 10 animals, for each group. For statistical significance were considered values *p* < 0.05. ^a^ Statistical significance in relation to CTRL group; ^b^ Statistical significance in relation to SAE group; ^c^ Statistical significance in relation to МetS + HFd group; ^d^ Statistical significance in relation to МetS + HFd + SAE group; ^e^ Statistical significance in relation to МetS + Sd group.

**Figure 5 ijms-20-00006-f005:**
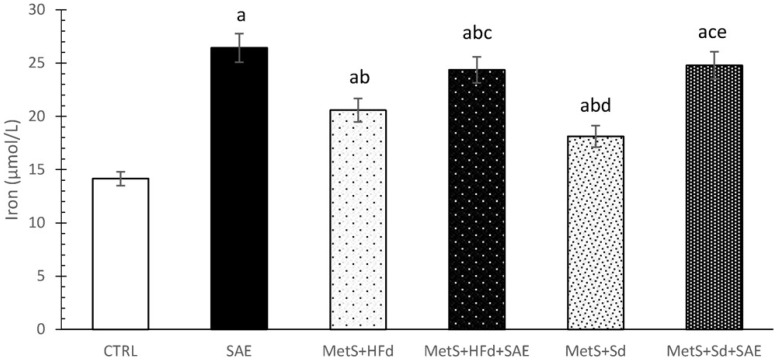
Effect of dietary changes in healthy and rats with MetS on serum iron levels. Values are expressed as mean ± standard deviation for 10 animals, for each group. For statistical significance were considered values *p* < 0.05. ^a^ Statistical significance in relation to CTRL group; ^b^ Statistical significance in relation to SAE group; ^c^ Statistical significance in relation to МetS + HFd group; ^d^ Statistical significance in relation to МetS + HFd + SAE group; ^e^ Statistical significance in relation to МetS + Sd group.

**Figure 6 ijms-20-00006-f006:**
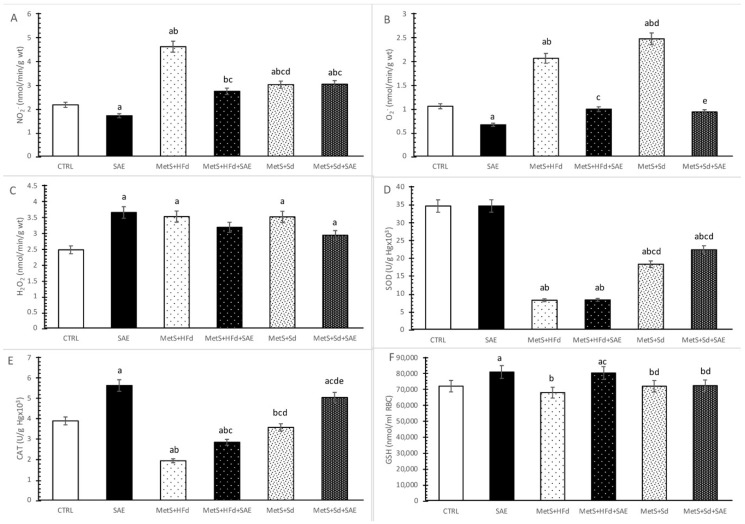
Effect of dietary changes in healthy and rats with MetS on systemic oxidative stress parameters: (**A**) NO_2_^−^; (**B**) O_2_^−^; (**C**) H_2_O_2_; (**D**) SOD; (**E**) CAT; (**F**) GSH. Values are expressed as mean ± standard deviation for 10 animals, for each group. For statistical significance were considered values *p* < 0.05. ^a^ Statistical significance in relation to CTRL group; ^b^ Statistical significance in relation to SAE group; ^c^ Statistical significance in relation to МetS + HFd group; ^d^ Statistical significance in relation to МetS + HFd + SAE group; ^e^ Statistical significance in relation to МetS + Sd group.

**Figure 7 ijms-20-00006-f007:**
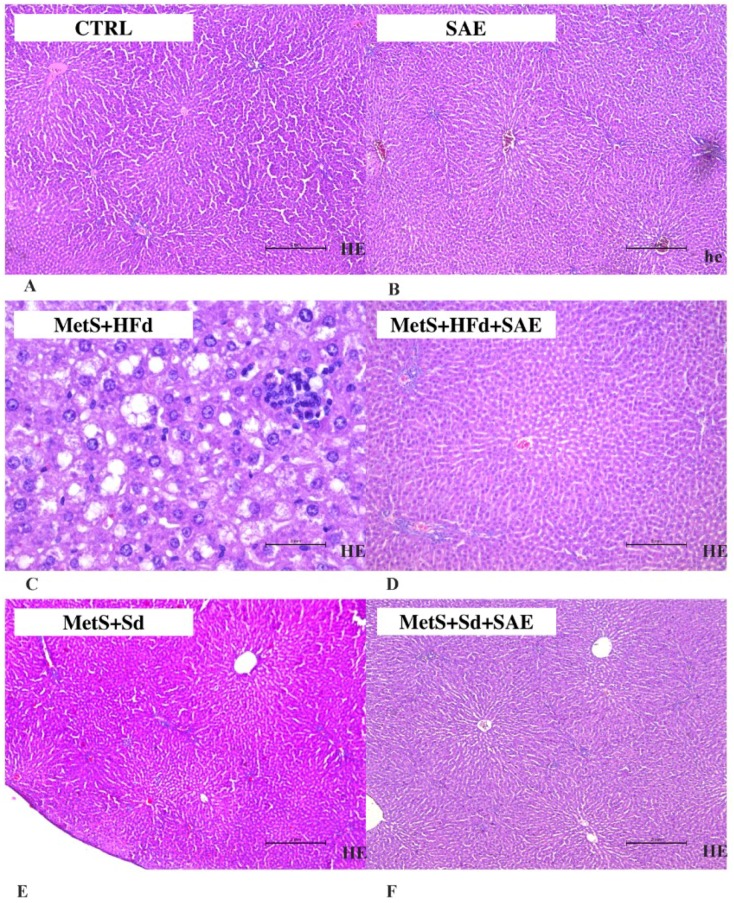
Representative hematoxylin and eosin (H&E) staining photos of liver tissue (for sub-figures A, B, D, E and F original magnification is 100× and scale bar is 2 mm, while in sub-figure C magnification is 400× and scale bar is 2 mm) in: (**A**) CTRL, (**B**) SAE, (**C**) МetS + HFd, (**D**) МetS + HFd + SAE, (**E**) МetS + Sd and (**F**) МetS + Sd + SAE groups.

**Table 1 ijms-20-00006-t001:** Effect of dietary changes in healthy and rats with MetS on in vivo cardiac function.

Milimeters (mm)	CTRL	SAE	MetS + HFd	MetS + HFd + SAE	MetS + Sd	MetS + Sd + SAE
IVSd	1.22 ± 0.2	1.89 ± 0.2 ^a^	1.28 ± 0.3 ^b^	1.34 ± 0.2	1.58 ± 0.2	1.59 ± 0.2
LVIDd	6.80 ± 0.4	5.73 ± 0.2 ^a^	6.23 ± 0.4	6.70 ± 0.4	5.53 ± 0.3	7.93 ± 0.4 ^e^
LVPWd	1.94 ± 0.4	3.45 ± 0.4 ^a^	2.22 ± 0.1	1.72 ± 0.1^b^	1.88 ± 0.2 ^b^	1.93 ± 0.2 ^b^
IVSs	2.48 ± 0.3	3.64 ± 0.3 ^a^	2.33 ± 0.1 ^b^	2.52 ± 0.2	2.45 ± 0.2 ^b^	3.24 ± 0.1 ^ace^
LVIDs	3.20 ± 0.5	2.03 ± 0.2 ^a^	3.24 ± 0.2 ^b^	3.16 ± 0.1	2.48 ± 0.2 ^acd^	3.45 ± 0.2 ^be^
LVPWs	2.98 ± 0.2	4.97 ± 0.1 ^a^	3.16 ± 0.2	3.26 ± 0.2	2.76 ± 0.2 ^b^	3.45 ± 0.3
FS (%)	53.2 ± 4.18	65.7 ± 5.01 ^a^	49.0 ± 3.99 ^ab^	52.2 ± 4.32^b^	51.4 ± 4.19 ^b^	56.3 ± 4.21

Values are expressed as mean ± standard deviation for 10 animals, for each group. For statistical significance were considered values *p* < 0.05. ^a^ Statistical significance in relation to CTRL group; ^b^ Statistical significance in relation to SAE group; ^c^ Statistical significance in relation to МetS + HFd group; ^d^ Statistical significance in relation to МetS + HFd + SAE group; ^e^ Statistical significance in relation to МetS + Sd group.

**Table 2 ijms-20-00006-t002:** Percentage differences between PCP 1 and PCP 2 during ex vivo perfusion on Langendorff aparatus.

CPP	CTRL	SAE	MetS + HFd	MetS + HFd + SAE	MetS + Sd	MetS + Sd + SAE
	dp/dt max					
60	−11.05	−5.99	−24.77	−1.07	−24.23	−13.13
80	−6.11	−12.90	−35.30	−4.41	−28.11	−3.77
100	−10.88	−4.79	−6.22	−4.08	−8.80	−14.02
120	−0.63	−1.91	5.62	−9.39	−2.06	−3.24
	dp/dt min					
60	−21.25	−14.81	−0.46	−0.62	−13.64	−6.36
80	−22.03	−11.53	−0.80	−9.03	−18.90	−2.32
100	−29.70	−11.43	−7.04	−3.81	−9.14	−12.42
120	−12.04	−8.14	−8.08	−3.15	2.63	1.16
	SLVP					
60	−8.35	−9.49	5.37	10.80	−10.58	4.06
80	−13.11	−1.66	5.70	8.55	−4.18	2.44
100	−16.81	−5.81	−1.22	2.15	3.72	−12.46
120	−7.79	−16.29	12.60	1.99	4.12	−0.07
	DLVP					
60	12.50	16.67	2.50	−15.79	14.41	6.67
80	16.05	10.20	−2.22	−19.05	9.32	4.76
100	10.47	−0.93	−4.21	−15.38	15.87	9.09
120	7.61	−5.26	0.00	−2.86	4.13	8.45
	HR					
60	−5.29	−4.01	−6.26	1.59	1.39	2.51
80	−1.96	−3.85	−6.82	0.23	0.41	1.10
100	−4.58	−2.77	−5.04	3.66	0.31	−0.37
120	−3.41	−2.63	−3.21	7.85	1.02	0.61
	CF					
60	−12.37	−10.07	−3.11	6.74	1.93	−4.55
80	−3.57	−10.40	−5.78	5.48	7.88	−3.36
100	−5.10	−8.33	−2.55	−0.57	13.44	−4.55
120	−1.21	−14.66	−14.96	1.51	7.59	−1.04

**Table 3 ijms-20-00006-t003:** Effect of dietary changes in healthy and rats with MetS on insulin levels during OGTT.

Groups	0′	180′
CTRL	122.9 ± 6.76	123.7 ± 6.61
SAE	106.9 ± 6.04 ^a^	113.8 ± 6.51
МetS + HFd	185.1 ± 7.78 ^ab^	129.3 ± 6.38
МetS + HFd + SАE	180.3 ± 8.02 ^ab^	131.1 ± 7.13 ^b^
МetS + Sd	205.8 ± 9.87 ^abcd^	145.2 ± 7.65 ^b^
МetS + Sd + SАE	182.1 ± 9.32 ^abcde^	127.3 ± 6.72 ^e^

Values are expressed as mean ± standard deviation for 10 animals, for each group. For statistical significance were considered values *p* < 0.05. ^a^ Statistical significance in relation to CTRL group; ^b^ Statistical significance in relation to SAE group; ^c^ Statistical significance in relation to МetS + HFd group; ^d^ Statistical significance in relation to МetS + HFd + SAE group; ^e^ Statistical significance in relation to МetS + Sd group.

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
