# Peer review of "Standardized Aronia melanocarpa Extract as Novel Supplement against Metabolic Syndrome: A Rat Model"

_ijms, 2018, doi:10.3390/ijms20010006_

Reviewer 1 Report

Title: Standardized Aronia melanocarpa Extract as Novel Supplement against Metabolic Syndrome: a Rat Model

Summary:

This is a rat model experiment on the effect of SAE against metabolic syndrome. Rats were randomly divided into 6 groups.  Cardiac function, liver morphology, and multiple blood biomarkers were compared across groups. The results showed promising evidence for the beneficial effects of SAE against MetS in multiple aspects. The major comments and suggestions are as follow:

1.      Please provide more information about the reasoning behind choosing current dose of SAE for your experiment. Is there any dose response evidence for SAE’s beneficial effects?

2.      Line 235, authors stated: “However, there was no change in HR when SAE was added to a dietary regimen both in rats with MetS who were on a high-fat or standard diet.” However, the figure 2.c showed that the heart rate for MetS+HFd+SAE is significantly higher than MetS+HFd.

Author Response

1. The recommended manufacturer dose of SAE for humans is 30 ml per day. We adjusted mention dose to rats’ body weight in our research.

2. Thank you for this useful observation. We corrected according to reviewer suggestion, please see Discussion section.

Reviewer 2 Report

The current study may be a starting point for further experimental and clinical investigations which would fully evaluate the effects of SAE alone or in combination with different dietary protocols in various models of chronic diseases.

Several epidemiologic studies have implicated visceral fat as a major risk factor for insulin resistance, type 2 diabetes mellitus, cardiovascular disease, stroke, metabolic syndrome and death [1]. Some results show beneficial effects of AM against hepatic lipid accumulation through the inhibition of PPARγ2 expression along with improvements in body weight, liver functions, lipid profiles and antioxidant capacity suggesting the potential therapeutic efficacy of AM on NAFLD [2]. Recently, it was showed clearly an increase in acetylcholinesterase (AChE) and butyrylcholinesterase (BChE) activity and disruption of lipid metabolism in patients with MetS. After supplementation of MetS patients with A. melanocarpa extract, a decrease in AChE activity and oxidative stress was noted [3].

Authors are kindly requested to emphasize the current concepts about these issues in the context of recent knowledge and the available literature. This article should be quoted in the References list.

References

Should visceral fat be reduced to increase longevity? Ageing Res Rev. 2013 Sep; 12 (4): 996-1004. doi:10.1016/j.arr.2013.05.007.

Aronia melanocarpa Extract Ameliorates Hepatic Lipid Metabolism through PPARγ2 Downregulation. PLoS One. 2017 Jan 12; 12 (1): e0169685. doi: 10.1371/journal.pone.0169685.

Changes in Cholinesterase Activity in Blood of Adolescent with Metabolic Syndrome after Supplementation with Extract from Aronia melanocarpa. Biomed Res Int. 2018 Mar 26; 2018: 5670145. doi: 10.1155/2018/56

Author Response

Thank you for such a useful suggestion. We inserted this paragraph in the Discussion section. This additional information regarding the mechanisms underlying beneficial effects of Aronia melanocarpa extract certainly improved the quality of the manuscript. Please see Discussion and References section.